# Development and Performance Evaluation of UHPC and HPC Using Eco-Friendly Additions as Substitute Cementitious Materials with Low Cost

**Mohammed Qusay Abdul Sahib [1], Masood Farzam [1,*] and Khalid A. Sukkar [2]**

[1] Faculty of Civil Engineering, University of Tabriz, Tabriz 51666-16471, Iran; qqmohammed76@gmail.com
[2] Department of Chemical Engineering, University of Technology-Iraq, Baghdad 10066, Iraq; khalid.a.sukkar@uotechnology.edu.iq
[*] Correspondence: mafarzam@tabrizu.ac.ir

**Abstract:** Ultra-high-performance concrete (UHPC) and high-performance concrete (HPC) are widely used in construction engineering applications. The quality and economy of this type of concrete are the main challenges in real construction systems due to their expensive cost. In the present investigation, the performances of UHPC and HPC were improved using eco-friendly additives from natural sources or industrial wastes. Accordingly, different kinds of concrete mixtures were prepared with the addition of various eco-friendly materials, such as metakaolin (10, 15, and 20%), silica fume (2.5, 5, 10, and 15%), cement kiln dust (CKD) (0, 5, and 10%), and 1 vol.% of steel and polypropylene fibers. All of these materials were subjected to efficient treatment and purification processes. The results indicated that the prepared UHPC was characterized by high compression and flexural strengths. The prepared UHPC (sample CR-2) with metakaolin (10%), CKD (10%), and 1 vol.% of steel fibers provided the highest compressive strength of 135 MPa at 28 days. Moreover, the results showed that reducing the cement amounts to 750, 500, and 250 kg/m$^3$ provided concrete with efficient structural requirements and specifications and can be characterized as UHPC and HPC. Also, the mixture (sample CM15) with a metakaolin addition of 15%, CKD of 100 kg/m$^3$, and 1 vol.% of steel fibers showed the highest flexural strength of 19.14 MPa at 28 d. Moreover, the highest splitting tensile strength of the prepared UHPC cylinders was 9.6 MPa at 28 d for the MSS1000 sample, which consisted of 15% metakaolin, a cement content of 1000 kg/m$^3$, silica fume of 10%, and steel fibers of 1% vol. The prepared UHPC mixtures will reduce the amount of consumed cement and the production cost, with a high performance in comparison to classical concrete. Finally, it was clear that the prepared UHPC and HPC concrete with green additions can serve efficiently in specific construction applications, with high performance, economic feasibility, and safe environmental impacts.

**Keywords:** high-quality concrete; green additives; tensile test; compression test; bending test; concrete cost





## 1. Introduction

In construction engineering, two advanced categories of concrete were designed to resist high load conditions. The first one is high-performance concrete (HPC), which is usually designed to give a compressive strength of more than 50 MPa, with high durability [1]. The second category is ultra-high-performance concrete (UHPC), which is also a cementitious composite material frequently employed in construction. Usually, such material is characterized by high tensile ductility, high compressive strength (greater than 120 MPa), high toughness, and specific durability [1–3]. The cost of a concrete member or structure is highly dependent on its structural dimensions [4]. Therefore, the use of UHPC and HPC structural members will clearly reduce the dimensions of the concrete structure. Moreover, the high mechanical properties of UHPC will reduce the number of sections, lower the requirements for passive reinforcing, and allow for a special structural design for

advanced applications [5–7]. Furthermore, UHPCs are highly durable in terms of resisting carbonation, abrasion, corrosion, and firing problems [6]. These structures usually perform well in terms of strength and the ability to deal with vibration and high seismic ratings, as well as providing environmentally safe materials [8–15].

Generally, concrete undergoes some operating problems in terms of workability and flow characteristics [16–20]. However, the addition of fly ash or silica and mineral materials can enhance the workability and flow specifications of a concrete mixture. The addition of these materials will reduce shrinkage in a concrete structure and enhance the strength and hardening of the UHPC and HPC [21–26]. Also, such materials can increase the concrete's resistance against sulfate attack problems and the extension of alkali-aggregate activity [19,27]. Usually, the production of UHPC needs a relatively high quantity of cement, and often, the required amount is not evaluated or optimized. Thus, the economic cost and environmental issues present the main challenges in the production process of UHPC [28–33]. Moreover, the use of UHPC in construction is limited due to its high cost of production. Therefore, there is a real need to increase the performance of concrete using sustainable materials. On the other hand, it would be advantageous to substitute some cheap materials from natural sources with a high performance into cement production [4,12,34], as this could extend the uses of UHPC to many construction engineering projects under severe conditions. This can be achieved by accurately managing different additions to the UHPC composition [35–38].

Many studies have investigated ways to improve the performance of UHPC and HPC using different kinds of additives. Christ et al. [39] reinforced the composition of UHPC with steel and polypropylene fibers. They used percentages of 50–100% and 0–50% of steel and polypropylene fibers, respectively, and found that the compressive strength reached 180 MPa with the additions of 80% steel fibers and 20% polypropylene fibers. Abubakar et al. [40] statistically evaluated the performance of HPC reinforced with single and hybrid fibers of steel, glass, and carbon, finding that the carbon fibers provided the best compression strength, while the steel fibers provided the best tensile strength. The authors also noted that the highest mechanical behaviors were achieved using single fibers and mixed fibers of 1 wt%. Qian et al. [41] used recycled concrete as an additive to synthesize UHPC green concrete, where recycled concrete powder was substituted for a percentage of more than 25% of the cement in the UHPC structure. Such additions of eco-friendly materials contributed clearly to enhancing the environment and specifications of concrete. Azmee et al. [42] prepared UHPC for the requirements of many construction applications and indicated that UHPC needed a high cement content. Accordingly, they studied the process of lowering the cement content in the UHPC using fly ash and ultrafine $CaCO_3$. The authors noted that the prepared low-cement UHPC was efficient and had a high strength after replacing up to 50% of the normal cement. Ozolins et al. [43] added micro-silica and weight of binder (wtob) to UHPC with percentages of 0–110% and 0–13.75%, respectively, noting that the compressive strength improved by 20% with a micro-silica dosage rate of 3.75% by binder.

The substitutions of some additives, such as fibers, micro-silica, glass, waste concrete, and fly ash, are key factors in the cost of the final prepared UHPC. Managing the process of such additions requires a deep understanding related to the structural operating mechanism of reinforced UHPC. Tagnit-Hamou et al. [44] added fine waste glass to UHPC as a supplementary material for cement and observed that such an addition reduced the operating costs of UHPC and enhanced the rheological specifications of the fresh concrete due to the replacement of cement with fine glass. El-Din et al. [45] improved the mechanical specifications of UHPC using additions of short and long steel fibers, and found that the UHPC provided high mechanical behaviors for all of these materials. They found that an increase in fiber content of 3% was optimal in all mechanical tests. Zhang et al. [46] evaluated the specification of alkali-activated A-UHPC reinforced with recycled concrete fines as a substitution material for cement. They indicated that the A-UHPC can provide the highest hardened behavior at a water–binder ratio of 0.27–0.29 and recycled concrete

content of 10–30% [46]. Abdal et al. [47] studied the main structural factors in UHPC to be used efficiently in applications of bridge engineering and showed their review on the interaction between many construction parameters and their influences on the final product cost.

The major benefits of the addition of green materials to concrete are reducing the cement amount, reducing environmental pollution, producing efficient concrete for high stress loads, and lowering the economic cost. Usually, green additions to UHPC include fine sand, quartz, metakaolin, and silica fume [22,36]. Other materials may be also added to UHPC, such as fly ash, waste concrete, waste glass, polymers (polypropylene), and stainless-steel wires [47]. It was noted from previous studies that the addition of various materials and green materials into a UHPC concrete mixture provides the advantage of improving the mechanical and structural specifications of the concrete. Therefore, the present investigation focused on using cheap and eco-friendly additives to produce UHPC and HPC. Such concrete can work efficiently in structural designs with reasonable costs. On the other hand, understanding the mechanism of using these alternative additives requires extensive study to understand their operating mechanism on the concrete structure. Also, it is required to evaluate the advantages, disadvantages, and weight percentage of each added material. Kuzielová et al. [48] studied the substitution of Portland cement using an eco-friendly material (silica fume at a weight% of 30 and 50) under a high formation temperature and hydrothermal conditions. The authors noted from the characterization results that the molecular water and the polymerization reaction were changed dramatically due to transformation reactions and then the crystallization of different phases in the concrete structure.

According to the literature review, there is a real need to provide more understanding of the influences of different kinds of additives to enhance the strength of UHPC and HPC. Also, UHPC and HPC blended with other materials will provide efficient crack resistance and can inhibit crack propagation, thereby reducing maintenance costs [49–56]. Typically, the cement content required in UHPC is high. However, there is a real need to reduce the amount of consumed cement and improve the performance of concrete in real construction applications. Therefore, the main aim of the present investigation was to prepare an efficient UHPC and HPC using green and cheap additives to provide high-performance concrete at a low cost.

## 2. Materials and Methods

### 2.1. Predominantly Added Eco-Friendly Materials

2.1.1. Pretreatment of White Kaolin Rocks

In the present work, white kaolin rocks were collected from western Iraq and subjected to two pretreatment steps. First, they were milled and ground to reduce their grain size to 150 μm using an industrial milling machine (Alpine Milling Machine, B500, Leingarten, Germany), as shown in Figure 1. Then, the kaolin powders were heated in an electric furnace at a calcination temperature of 750 °C for 1.5 h. Figure 2 displays a photograph of the calcination furnace. After the heating process, the samples were brought down to room temperature at ambient conditions to avoid crystallization of the amorphous metakaolin, as shown in Figure 2b. Usually, the calcination process is a key step in activating the alumina layer within the kaolin structure, which enhances its reaction with acid. The weights of the samples before and after the thermal treatment were measured to determine the weight loss during the calcination process. Metakaolin consists of alumina, silica, and some impurities, but all of the impurities disappeared by loss on ignition (LOI) due to the calcination process. Moreover, organic materials can be removed at a temperature of 230 °C. Table 1 summarizes the chemical analysis of the produced metakaolin.

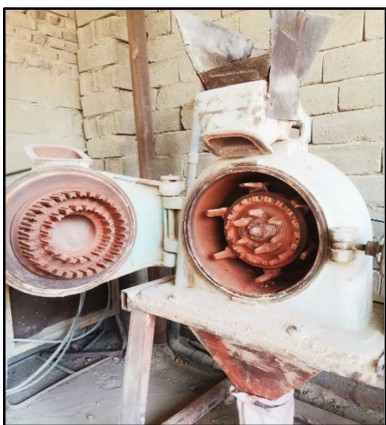

**Figure 1.** Milling machine for white kaolin rocks.

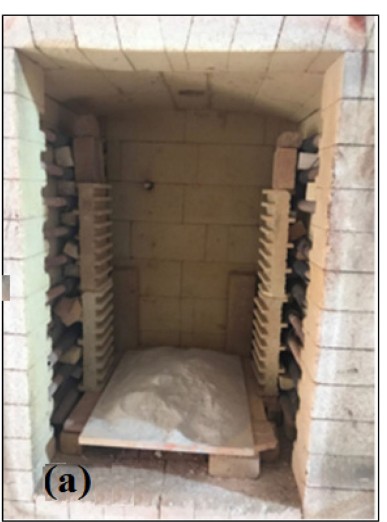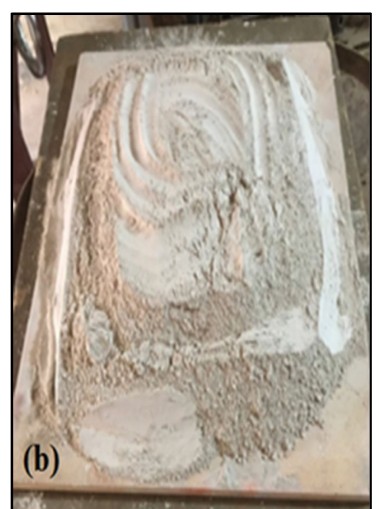

**Figure 2.** Calcination process of kaolin powders: (**a**) electric furnace at 750 °C for 1.5 h, and (**b**) metakaolin produced via the calcination process.

**Table 1.** Chemical composition of the produced natural metakaolin.

| Oxide Type | $Al_2O_3$ | $SiO_2$ | $Fe_2O_3$ | $K_2O$ | $TiO_2$ | CaO | $SO_3$ | $V_2O_5$ | $Cr_2O_3$ | $Ta_2O_3$ | SrO | MnO | $Ir_2O_3$ | $ZrO_2$ |
|---|---|---|---|---|---|---|---|---|---|---|---|---|---|---|
| wt% | 61.852 | 35.55 | 0.895 | 0.890 | 0.516 | 0.205 | 0.027 | 0.025 | 0.015 | 0.013 | 0.002 | 0.002 | 0.002 | 0.008 |

According to ASTM standard C618-17, the calcination of natural clay can be categorized as a pozzolanic material if the summation of the three main oxides ($SiO_2$, $Al_2O_3$, and $Fe_2O_3$) is equal to or higher than 70%. It can be concluded that all Iraqi metakaolin can be classified as pozzolanic for use in concrete.

### 2.1.2. Pretreatment of White Quartz

High-purity silica sand (quartz) was collected from the Al-Ramadi western region of Iraq. This material was subjected to a washing process with deionized water to remove dust and impurities, and then dried in an electric furnace at a temperature of 110 °C for 6 h. The dried products were ground into powders using a grinding machine for 4 h to reduce the grain size to 150 μm. This quartz material was characterized by its high content of $SiO_2$ (95.652%). Accordingly, such material is regarded as an effective, low-cost, green material. Table 2 shows the chemical composition of the Iraqi quartz after the milling process.

**Table 2.** Chemical analysis of the milled quartz after 4 h of milling.

| Oxide Type | Al$_2$O$_3$ | SiO$_2$ | Fe$_2$O$_3$ | K$_2$O | TiO$_2$ | CaO | SO$_3$ | CuO | MnO |
|---|---|---|---|---|---|---|---|---|---|
| wt% | 3.888 | 95.652 | 0.133 | 0.157 | 0.007 | 0.115 | 0.035 | 0.004 | 0.009 |

### 2.1.3. Pretreatment of Cement Kiln Dust (CKD)

Cement kiln dust (CKD) is fine-grained particulate dust collected from electrostatic precipitators during cement production under high temperatures. The chemical composition of CKD is similar in terms of calcium carbonate and silica oxide to that of the raw materials used in feeding kilns to make ordinary Portland cement. The cement kiln dust (CKD) was obtained from the Kufa Cement Plant (Al-Najaf City, Iraq) and used in this experimental work as an active additive. Figure 3 shows a photograph of the cement plant. The relatively high alkaline content of CKD is the predominant factor preventing its recycling in cement manufacturing. One effective way to utilize CKD is to use blended cement as a partial replacement for cement in UHPC structure. Table 3 illustrates the chemical composition of the CKD that was used.

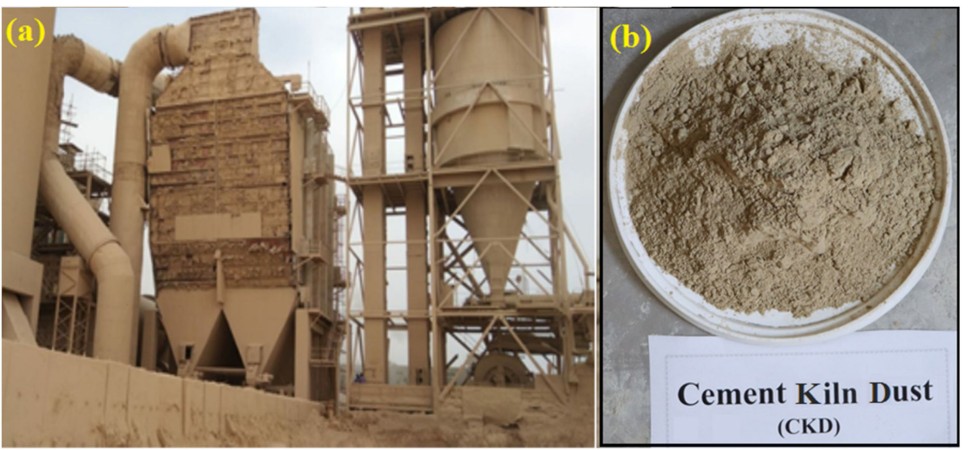

**Figure 3.** The collection process of CKD material: (**a**) the cement plant site and (**b**) final CKD.

**Table 3.** Chemical analysis of Iraqi CKD.

| Oxide Type | CaO | SiO$_2$ | Al$_2$O$_3$ | Fe$_2$O | SO$_3$ | MgO | LOI |
|---|---|---|---|---|---|---|---|
| wt% | 48.37 | 12.02 | 2.91 | 2.78 | 7.10 | 2.04 | 24.78 |

### 2.1.4. Steel Wire Rope Waste

Steel fibers are widely employed in UHPC to enhance their ability to resist high stress. In the present work, steel wire rope waste was used as a new kind of alternative additive to UHPC. Such material is characterized by low cost and high load ability. To improve the UHPC's specifications inexpensively, these steel wires were added as a reinforcing material to replace or partially replace traditional steel and synthetic fibers. Replacing steel fibers decreases the amount of steel wire waste and greatly contributes to the reduction of environmental pollution.

The experimental work included cutting the collected steel wire rope waste into small pieces, 20 mm in length and 0.2 mm in diameter. A high-force cutting tool (i.e., hydraulic wire rope cutter, Hi-Force, HSWC28, Daventry, UK) was employed for the cutting processes, as shown in Figure 4.

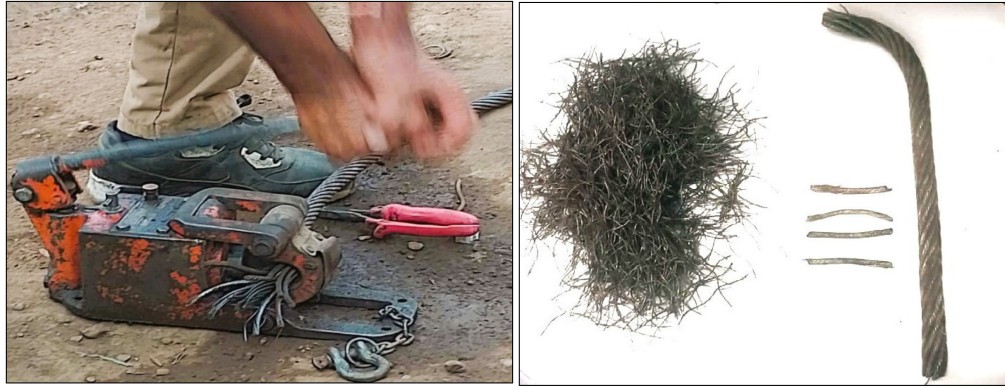

**Figure 4.** Collected steel wires from steel rope waste after the cutting processes.

### 2.1.5. Polypropylene Woven Fibers (Food Bag Waste)

Polypropylene fabric is a textile made from polypropylene (PP), characterized by its thermoplastic properties and usually used as a packaging material for food, furniture, grains, films, automotive parts, and medical devices. In the present investigation, polypropylene woven bag waste was used as an additive to enhance the mechanical properties of UHPC. The fibers were simply cut from locally available polypropylene woven bag waste with sizes of up to 25 mm in fiber length, as shown in Figure 5. Table 4 summarizes the main properties of this PP. As such, the reuse of polypropylene waste material will reduce plastic environmental pollution while also enhancing the mechanical properties of the UHPC.

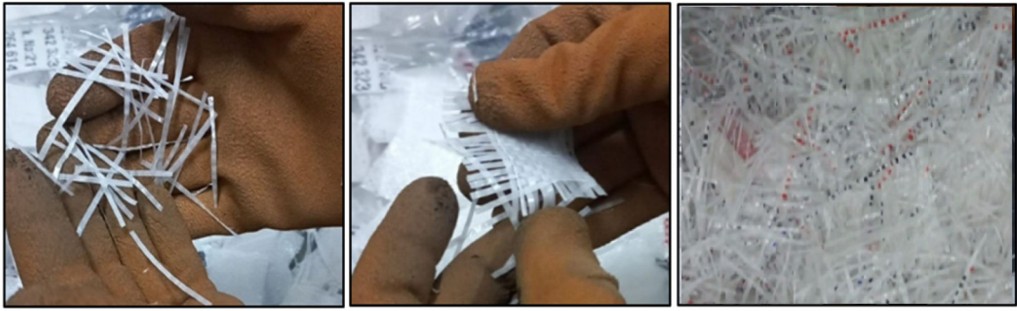

**Figure 5.** Production of polypropylene fibers from food bag waste by cutting the fibers into small pieces, 25 mm in length.

**Table 4.** General properties of polypropylene fibers.

| Property | Shape | Sp.gr (g/cm³) | Tensile Strength (GPa) | Elasticity (GPa) | Length (mm) | Diameter (µm) |
|---|---|---|---|---|---|---|
| Value | Longitudinal Fibers | 0.9 | 0.352 | 3.51 | 25 | 15 |

### 2.1.6. Micro-Sand

Natural sand was collected from the city of Al-Anbar, Iraq, and used in the concrete mixes. The sand was cleaned and washed using deionized water and air-dried for 24 h. The dry sand was subjected to a screening process followed by grinding by a machine. The produced micro-sand was very fine, with a maximum size of 150 µm. The chemical analysis results of the micro-sand are shown in Table 5.

**Table 5.** Chemical composition of the micro-sand.

| Oxide Type | $Al_2O_3$ | $SiO_2$ | $Fe_2O_3$ | $K_2O$ | $TiO_2$ | CaO | $SO_3$ | CuO | $ZrO_2$ | MnO | SrO | $GeO_2$ |
|---|---|---|---|---|---|---|---|---|---|---|---|---|
| wt% | 11.320 | 87.524 | 0.202 | 0.166 | 0.110 | 0.474 | 0.17 | 0.005 | 0.019 | 0.007 | 0.002 | 0.001 |

### 2.1.7. Superplasticizer (SP)

To improve the workability and strength of the concrete, a high-range water-reducing admixture (ViscoCrete-180G, Sika Company, Sydney, Australia) was added. This superplasticizer was obtained from Sika Company, Baar, Switzerland. The specifications of this material are shown in Table 6.

**Table 6.** Specifications of the superplasticizer, ViscoCrete®-180G.

| Property | Appearance | Sp.gr (g/cm$^3$) | pH | Chloride Content |
|---|---|---|---|---|
| Specification | Light-brownish liquid | 1.065 | 4–6 | Null |

### 2.2. Preparation Procedure of UHPC and HPC with Additions

UHPC and HPC samples were prepared according to standard procedures, by adding various weight percentages of local, alternative cementitious materials. Tables 7–10 summarize the prepared mixtures of UHPC using control samples, and additions of MK, STF, PPF, SF, MSD, QZ, and CKD.

**Table 7.** Preparation mixtures of UHPC.

| | Mix. No. | C kg/m$^3$ | CKD kg/m$^3$ | MK kg/m$^3$ | SF kg/m$^3$ | QZ kg/m$^3$ | MSD kg/m$^3$ | STF V% | PPFV% | SP kg/m$^3$ | W/B |
|---|---|---|---|---|---|---|---|---|---|---|---|
| | **Experimental Components of the Prepared Parent UHPC** | | | | | | | | | | |
| 1 | CR-1 | 1000 | 0 | 111 (10%) | 0 | 330 | 1100 | 1 | 0 | 25 | 0.32 |
| 2 | CR-2 | 1000 | 0 | 0 | 0 | 330 | 1100 | 1 | 0 | 25 | 0.32 |
| 2 | CR-3 | 900 | 0 | 100 (10%) | 0 | 330 | 770 | 1 | 0 | 25 | 0.32 |
| 3 | CR-4 | 850 | 0 | 150 (15%) | 0 | 330 | 770 | 1 | 0 | 25 | 0.32 |
| | **Experimental Components of the STF Additions** | | | | | | | | | | |
| 1 | CM10 | 750 | 100 | 95 (10%) | 0 | 330 | 770 | 1 | 0 | 25 | 0.32 |
| 2 | CM15 | 750 | 100 | 150 (15%) | 0 | 330 | 770 | 1 | 0 | 25 | 0.32 |
| 3 | CM20 | 750 | 100 | 212 (20%) | 0 | 330 | 770 | 1 | 0 | 25 | 0.32 |
| | **Experimental Components of PPF Additions** | | | | | | | | | | |
| 1 | CP10 | 750 | 100 | 95 (10%) | 0 | 330 | 770 | 0 | 1 | 25 | 0.32 |
| 2 | CP15 | 750 | 100 (10%) | 150 (15%) | 0 | 330 | 770 | 0 | 1 | 25 | 0.32 |
| 3 | CP20 | 750 | 100 (10%) | 212 (20%) | 0 | 330 | 770 | 0 | 1 | 25 | 0.32 |

**Table 8.** The prepared mixtures of the UHPC with MK and SF additions.

| | Mix No. | C kg/m$^3$ | CKD kg/m$^3$ | MK kg/m$^3$ | SF kg/m$^3$ | QZ kg/m$^3$ | MSD kg/m$^3$ | STF V% | PPFV% | SP kg/m$^3$ | W/B |
|---|---|---|---|---|---|---|---|---|---|---|---|
| | **Experimental Components of MK and SF Additions** | | | | | | | | | | |
| 1 | MSSP-1 | 750 | 0 | 100 (10%) | 150 | 330 | 770 | 1 | 1 | 25 | 0.32 |
| 2 | MSSP-2 | 750 | 0 | 150 (15%) | 100 | 330 | 770 | 1 | 1 | 25 | 0.32 |
| 3 | MSSP-3 | 750 | 0 | 200 (20%) | 50 | 330 | 770 | 1 | 1 | 25 | 0.32 |

**Table 9.** Prepared UHPC mixtures with MK, SF, CKD, and STF additions.

| | Mix No. | C kg/m³ | CKD kg/m³ | MK kg/m³ | SF kg/m³ | QZ kg/m³ | MSD kg/m³ | STF V% | PPFV% | SP kg/m³ | W/B |
|---|---|---|---|---|---|---|---|---|---|---|---|
| **Experimental Components of MK, SF, and CDK Additions** | | | | | | | | | | | |
| 1 | MSCKSP-1 | 750 | 50 (5%) | 100 (10%) | 100 (10%) | 330 | 770 | 1 | 1 | 25 | 0.32 |
| 2 | MSCKSP-2 | 750 | 50 (5%) | 150 (15%) | 50 (5%) | 330 | 770 | 1 | 1 | 25 | 0.32 |
| 3 | MSCKSP-3 | 725 | 50 (5%) | 200 (20%) | 25 (2.5%) | 330 | 770 | 1 | 1 | 25 | 0.32 |
| **Experimental Components of MK and SF Additions** | | | | | | | | | | | |
| 1 | MSS-1 | 750 | 0 | 100 (10%) | 150 (15%) | 330 | 770 | 1 | 0 | 25 | 0.32 |
| 2 | MSS-2 | 750 | 0 | 150 (15%) | 100 (10%) | 330 | 770 | 1 | 0 | 25 | 0.32 |
| 3 | MSS-3 | 750 | 0 | 200 (20%) | 50 (5%) | 330 | 770 | 1 | 0 | 25 | 0.32 |
| **Experimental Components of SF and STF Additions** | | | | | | | | | | | |
| 1 | SS-1 | 750 | 0 | 0 | 250 | 330 | 770 | 1 | 0 | 25 | 0.32 |
| 2 | SSp-1 | 750 | 0 | 0 | 250 | 330 | 770 | 1 | 1 | 25 | 0.32 |

**Table 10.** Prepared UHPC mixtures with MK and SF additions.

| | Mix No. | C kg/m³ | CKD kg/m³ | MK kg/m³ | SF kg/m³ | QZ kg/m³ | MSD kg/m³ | STF V% | PPFV% | SP kg/m³ | W/B |
|---|---|---|---|---|---|---|---|---|---|---|---|
| **Experimental Components of MK and SF Additions** | | | | | | | | | | | |
| 1 | MSS1000 | 1000 | 0 | 200 (15%) | 133 | 330 | 1100 | 1 | 0 | 25 | 0.32 |
| 2 | MSS750 | 750 | 0 | 100 (10%) | 150 | 330 | 1100 | 1 | 0 | 25 | 0.32 |
| 3 | MSS500 | 500 | 0 | 400 (40%) | 100 | 330 | 1100 | 1 | 0 | 25 | 0.32 |
| 4 | MSS250 | 250 | 0 | 650 (65%) | 100 (10%) | 330 | 1100 | 1 | 0 | 25 | 0.32 |

The mixing procedure was carried out by blending all of the dry cementitious materials (i.e., cement, SF, QZ, MSD, CKD, and MK) in a pan mixer for 1 min to obtain a homogenized powder. Next, 50 percent of both the water and the superplasticizer were added to the dry materials under a continuous mixing operation for 1 min. After that, the remaining water and the superplasticizer were added to the mixture for 0.5 min. Finally, the fibers (i.e., STF or PPF) were added gradually by hand to distribute them uniformly, and then, the mixing was continued for 5 min. Consequently, the mix was converted into a thick slurry paste. After a sufficient mixing time of 5 min, the mixer pan was stopped and the concrete was poured into either cubic molds (10 cm × 10 cm × 10 cm), cylinder molds (10 cm in diameter and 20 cm in height), or prism molds (4 cm × 4 cm × 14 cm). The samples were consolidated on a vibrating table to remove air from the concrete. All specimen molds were covered with plastic sheets to prevent moisture evaporation from the mortar during dry curing for 24 h at room temperature. Then, the test specimens were removed from the molds and immersed in a water curing tank at a temperature of 20 ± 2 °C to await testing after 3, 7, and 28 d. Figure 6 summarizes the preparation stages of the parent and modified UHPC with different additions, including mixing, processes of concrete contents, and the formation of cubes, cylinders, and prisms.

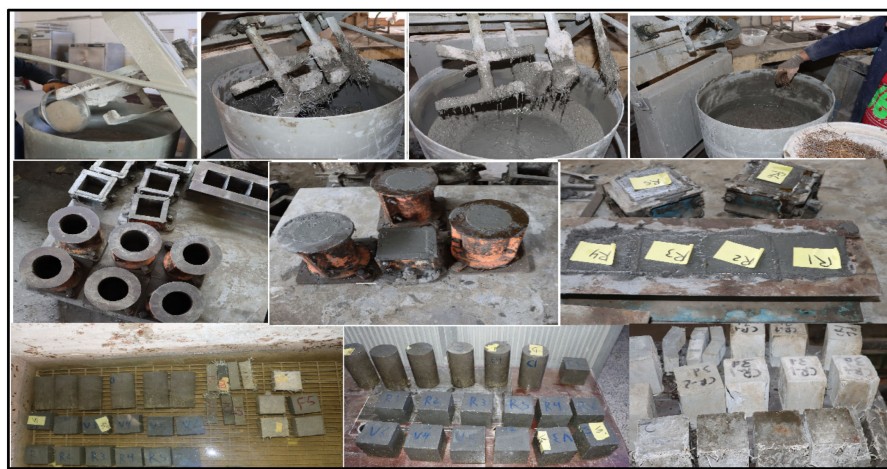

**Figure 6.** Main stages of the parent and modified UHPC preparations, including the curing process and the formation of cubes, cylinders, and prisms.

### 2.3. Testing Methods of Prepared UHPC and HPC

The mechanical properties of the prepared samples of the parent and modified UHPCs with various additives were measured using several testing methods. All UHPC and HPC specimens were tested in a compression testing machine (EN-12390-4 and EN-772-1 compression frames, Controls S.P.A., Milan, Italy). The machine's capacity was 1500 kN, as per ASTM C109 for cubes and ASTM C39 for cylinders. The compression tests were conducted on 160 cube specimens. For the 120 prisms with dimensions of 4 cm × 4 cm × 14 cm after 3, 7, and 28 d of water curing, the fracture properties were evaluated by conducting a standard three-point flexural loading test, as per ASTM C 78. Similarly, the split tensile strength tests on cylinders were conducted according to ASTM C496. Figure 7 shows the mechanical properties measured on the UHPC in the testing machine.

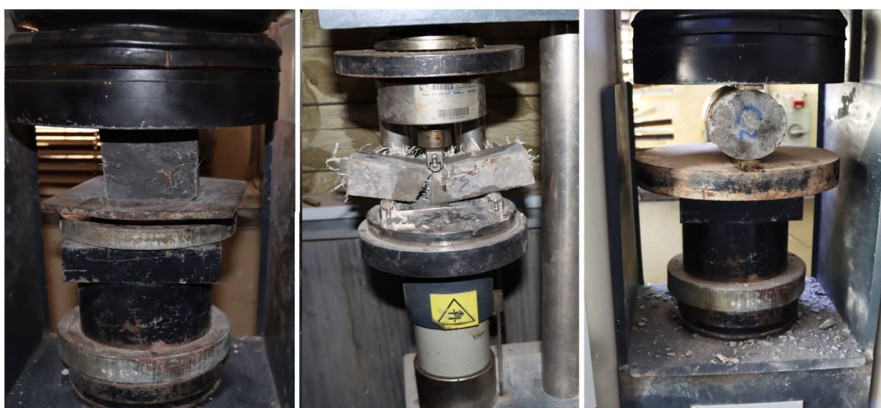

**Figure 7.** Testing machine for obtaining the mechanical properties of the parent and modified UHPC samples.

Accordingly, to assure the validity of the testing results, the prepared cubes of concrete were measured twice (two cubes of the same components) in the testing process. The errors of measurements were evaluated and then the uncertainty assessment was determined using the standard deviations between the different experimental measurements [51]. The SD% were from 2 to 3.6%. Also, all testing devices were efficiently calibrated to lower the practical errors. The uncertainty measurements were ±2.4, ±3.1, and ±3.6 for the compressive strengths, flexural strengths, and split tensile strengths, respectively.

## 3. Results and Discussion

### 3.1. Compressive Strength

For many structural engineering applications, compressive strength is the main criterion for concrete resistance evaluation under high stress. Figure 8 shows the comparison results of the compressive strengths of several UHPC mixtures at different additions of MK (10, 15, and 20%). These samples were reinforced with constant amounts of 10% CKD and 1 vol.% of steel fibers. This figure shows that specimen CR-2 provided the highest compressive strength of 135 MPa at 28 d. Moreover, the results indicated that the compressive strength decreased at all curing ages for sample CM20, which was prepared by adding 20% metakaolin and 10% CKD. The highest compressive strength achieved was 108.2 MPa. In contrast, the compressive strengths showed the values of 106.2 and 104.6 MPa for samples CM15 and CM20, respectively. It was observed that MK works as an efficient filler and enhances the pozzolanic process. Such an improvement will slow the release of lime while also increasing the hydration process (calcium silicate) in concrete. It is important to mention here that the results values of the compressive strengths indicated the production of HPC, not UHPC, because these values were less than the standard for UHPC (<120 MPs). Accordingly, the enhanced concrete specifications are regarded as a real contribution to construction by producing HPC from eco-friendly material additives.

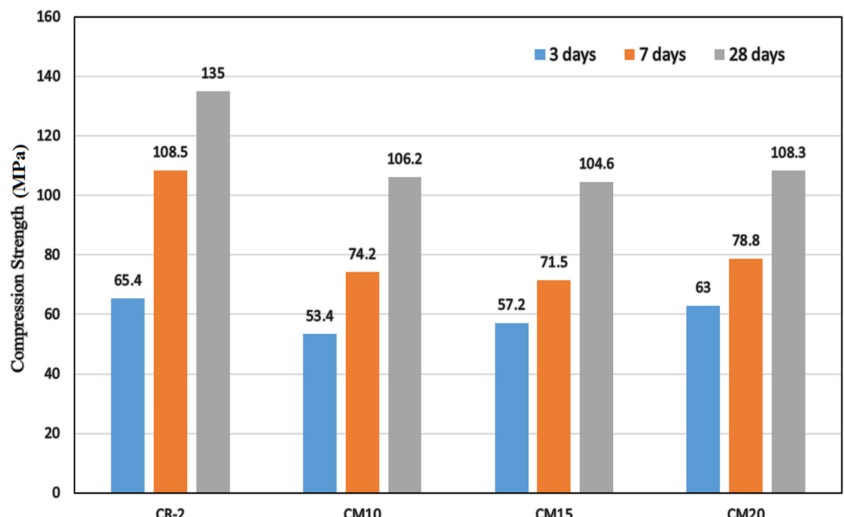

**Figure 8.** Compressive strengths of the prepared UHPC mixtures after 3, 7, and 28 d, including the partial replacements of cement by MK (10, 15, and 20%) at a constant CKD of 100 kg/m$^3$ and 1 vol.% of STF.

Figure 8 reveals that the maximum compressive strength was noted with an addition of 20 wt% metakaolin as an active replacement of the cement in the prepared UHPC. However, specimen CR-2 showed the optimal cement reduction of 25%. Additionally, the UHPC reinforced with waste rope wire steel fibers (STF) at 1 vol.% enhanced the mechanical properties of the prepared concrete dramatically. Such material is characterized by its low cost in comparison with that of imported steel fibers. Also, the use of waste steel rope wire fibers will lower the negative environmental impact due to the reuse of this material with high performance in the UHPC structure, in agreement with previous studies [7,18,25].

Figure 9 illustrates the effect of the addition of STF and PPF on the compressive strength of the prepared UHPC, using 1 vol.% of both fibers. All other materials in the concrete mixes were kept constant, as shown in Figure 8. The mixed concrete reinforced with the addition of both types of fibers had a lower compressive strength than that of specimen CR-2 for all curing ages. The addition of STF and PPF to the UHPC samples affected the deterioration of its mechanical specifications [23,40]. Sciarretta et al. [49] showed that UHPC required a hybrid mixture of STF and PPF to be used at high operating temperatures.

This can be attributed to the fact that strength can decrease in a hot environment, and these fibers provide the necessary reinforcement. Moreover, PPF alone cannot resist the occurrence of spalling in the UHPC structure, especially in hot environments [5,8,39].

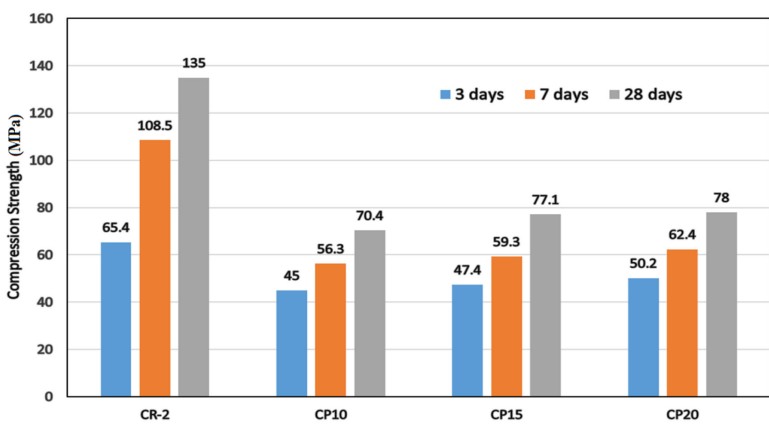

**Figure 9.** Compressive strengths of the prepared UHPC mixtures after 3, 7, and 28 d, including partial replacements of cement by various proportions of MK (i.e., 10, 15, and 20%) at a constant CKD of 100 kg/m$^3$ and 1 vol.% of PPF.

Comparing the results in Figures 8 and 9 demonstrates that the compressive strength was dramatically influenced by the additions of both types of fibers. The highest compressive strength of the mixed concretes was achieved with a water/binder ratio (W/B) of 0.32 [55,56] and a fiber volume fraction of 1 vol.%, with the values of 70.4, 77.1, and 78 MPa for CP10, CP15, and CP20, respectively, with the same observation noted by some previous studies [8,29,45]. Actually, such values of compressive strengths indicated the formation of HPC (compressive strength < 120 MPa). From a structural point of view, the predominant superior specifications of HPC and UHPC mainly resulted from the mix design, which contained fine silica, quartz, and a low water/binder ratio as well as a high cement content. All of these materials were added in the optimal ratios for the parent and modified UHPC samples.

To evaluate the influence of the additions of STF and PPF together on the performance of the prepared concrete, some samples were prepared for this purpose, as shown in Figure 10. The compressive strengths were measured in the presence of MK at 10, 15, and 20%, respectively, and silica fume at 15, 10, and 5%, with 1 vol.% of both steel and polypropylene fibers. The results indicated that substituting eco-friendly materials (i.e., MK, SF, STF, and PPF) into the concrete structure maintained the HPC's high quality. This was noted especially within two samples, MSSP-1 and MSSP-2, which provided the compressive strengths of 110.9 and 115.4 MPa, respectively. These results indicated that the addition of such eco-friendly materials contributed efficiently to the production cost of concrete (replacement of cement) with high performance.

Additionally, Figures 11–14 show the effects of different additions on the compressive strength. All of these samples indicated excellent improvement in the modified concrete under all ranges of additions. The results presented in Figure 13 indicate that the compressive strength of mixtures containing a partial replacement of 25% silica fume and 1 vol.% of STF reached the highest value of 126.9 MPa for sample SS-1. The presence of fibers contributes to enhancing the mechanical behavior of the UHPC. Furthermore, the addition of a high percentage of fibers produces low interfacial strength in the structure of UHPC. Therefore, in the present work, clear management of the STF and PPF additions was achieved to produce UHPC with high efficiency. Sun et al. [6], Shi et al. [25], Faried et al. [30], and El-Din et al. [45] have pointed out the importance of the management process of various additions in the mixture of UHPC to obtain efficient mechanical properties with a long lifetime. Such management meets the main challenge of reducing the amounts of cement and steel fibers while clearly enhancing the concrete specifications [15,47,49].

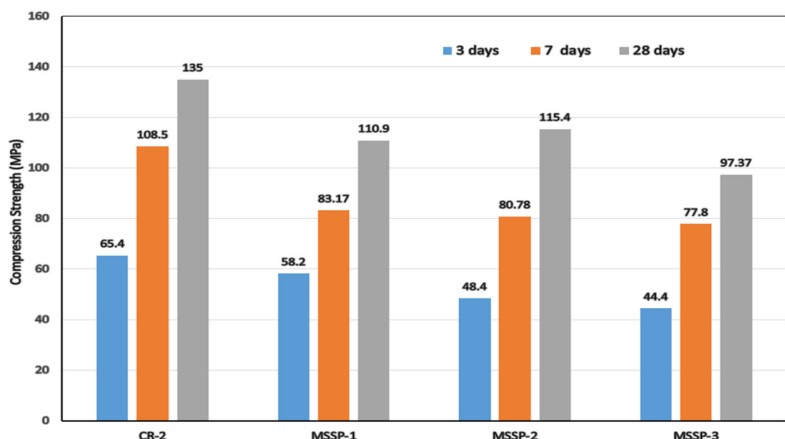

**Figure 10.** The compressive strengths of the mixtures containing partial replacements of MK (10, 15, and 20%), silica fume (15,10, and 5%), and 1 vol.% of STF and PPF at the curing times of 3, 7, and 28 days.

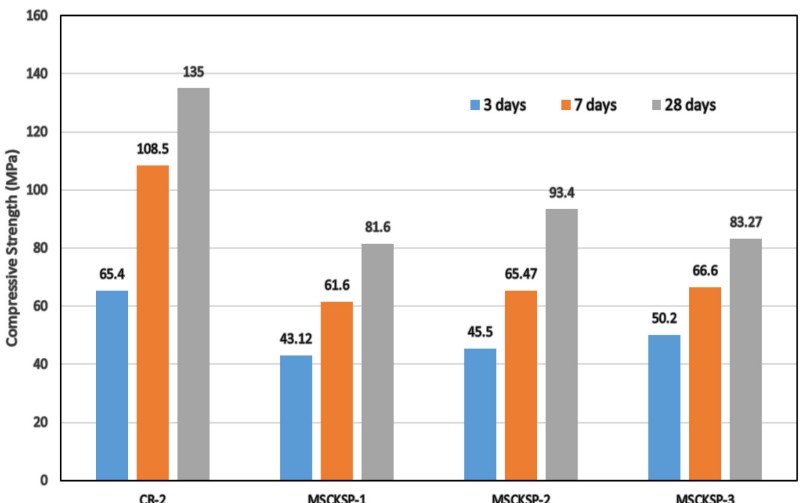

**Figure 11.** Compressive strengths of the mixtures containing partial replacements of MK (10, 15, and 20%), a constant CKD amount of 50 kg for each one $m^3$, silica fume (10, 5, and 2.5%), and 1% vol. of STF and PPF at the curing times of 3, 7, and 28 days.

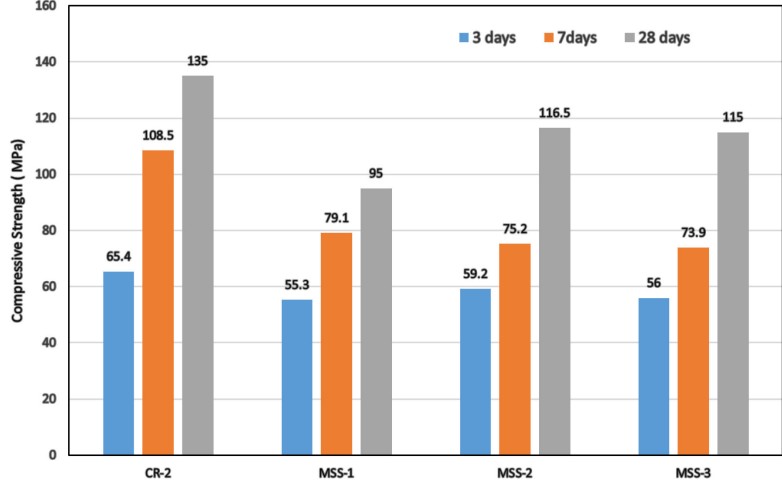

**Figure 12.** Compressive strengths of the mixtures containing partial replacements of MK (10, 15, and 20%), SF (15, 10, and 5%), and 1 vol.% of STF at curing times of 3, 7 and 28 days.

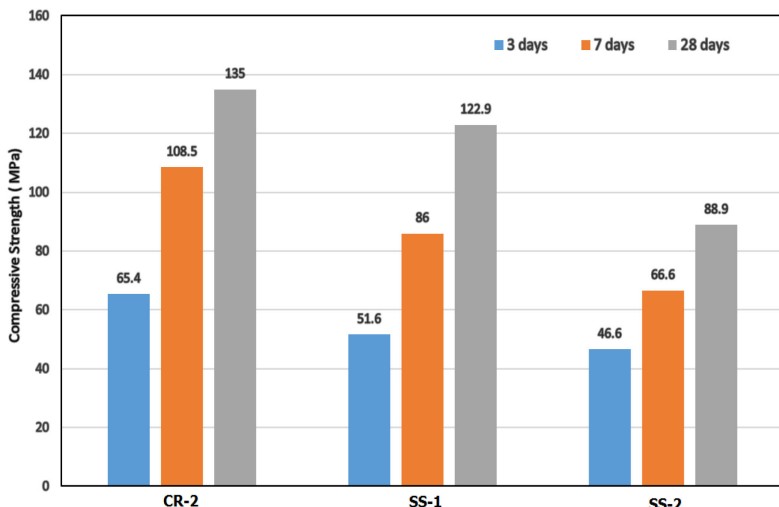

**Figure 13.** Compressive strengths of the mixtures containing partial replacements of a constant SF of 25% and 1 vol.% of STF at the curing times of 3, 7, and 28 days.

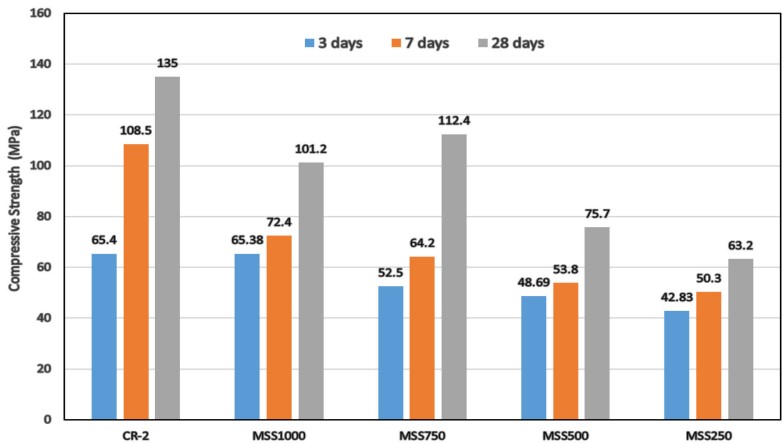

**Figure 14.** Compressive strengths of the mixtures containing different cement ratios and 1 vol.% of STF at the curing times of 3, 7, and 28 days.

### 3.2. Flexural Strength

The flexural strengths of the prepared UHPC mixtures of different eco-friendly additions (i.e., MK, SF, CDK, and STF, and PPF) are shown in Figures 15–20. Figure 15 displays the results of the flexural strengths of the mixed concrete with the addition of MK and STF (1 vol.%). After 28 days, the flexural strengths recorded values of 12.58, 19.14, and 17.81 MPa for the 10, 15, and 20% additions of MK, respectively. The MK additions clearly contributed to enhancing the mechanical properties of the prepared concrete. Such green material fills the internal pores of the concrete matrix efficiently [8,26]. In such cases, calcium silicate hydrate (C-S-H) gel is usually formed due to the pozzolanic reaction of metakaolin [19,26]. Then, the process combined $Ca(OH)_2$ into the mixture; as a result, the microstructure of the UHPC was enhanced. Also, Figure 15 demonstrates that the flexural strength achieved its highest value with the MK addition of 15% in sample CM15. Therefore, it was found that the substitution of MK for cement significantly improved the ductility and reduced the thickness of the transition zone in the UHPC matrix for all curing ages. The same observations were noted by Gong et al. [8], Abdellatief et al. [11], and Bahmani and Mostofinejad [19].

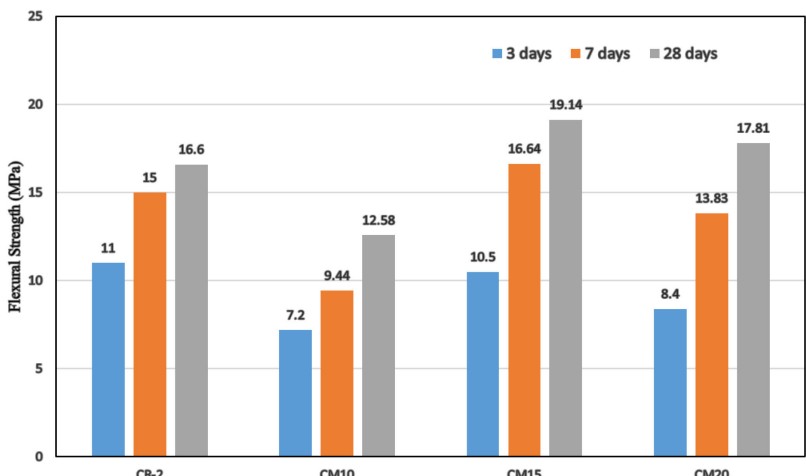

**Figure 15.** Flexural strengths of the mixtures containing partial replacements of MK (10, 15, and 20%) with a constant amount of CKD (100 kg/m$^3$), with 1 vol.% of STF at the curing times of 3, 7, and 28 days.

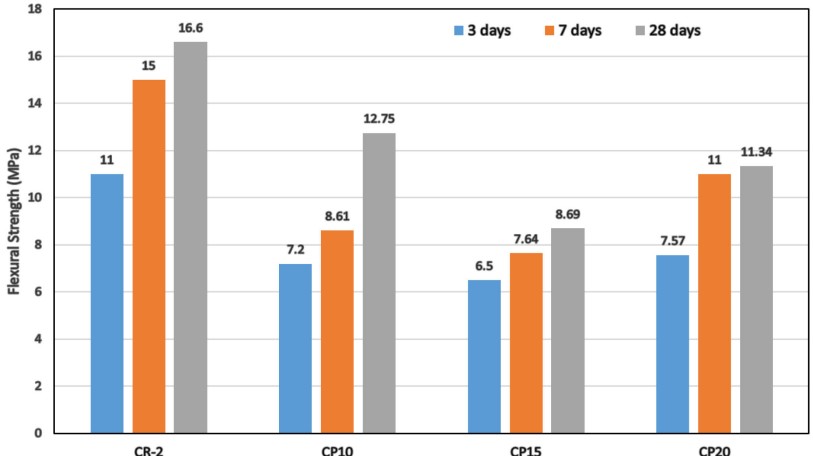

**Figure 16.** Flexural strengths of the mixtures containing partial replacements of MK (10, 15, and 20%), a constant amount of CKD (100 kg/m$^3$), and 1 vol.% of PPF at the curing times of 3, 7, and 28 days.

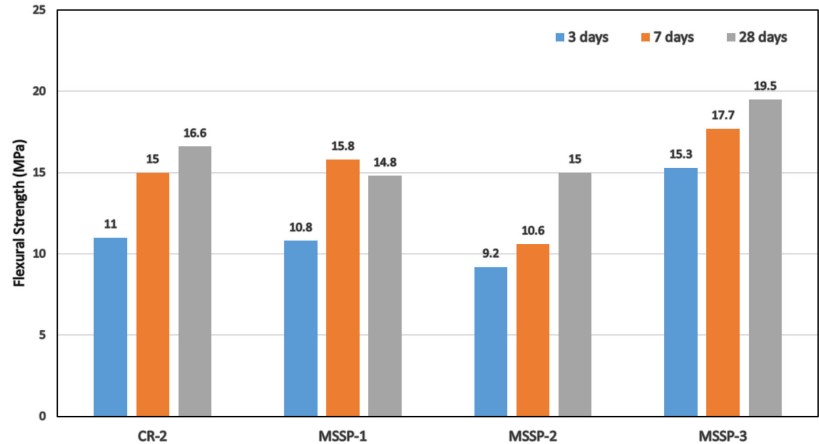

**Figure 17.** Flexural strengths of the mixtures containing partial replacements of MK (10, 15, and 20%), silica fume (15, 10, and 5%), and 1 vol.% STF and PPF at the curing times of 3, 7, and 28 days.

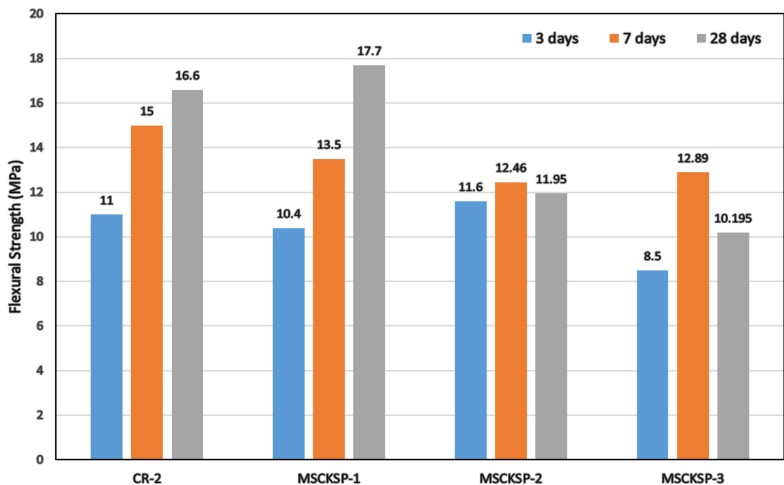

**Figure 18.** Flexural strengths of the mixtures containing partial replacements of MK (10, 15, and 20%), a constant amount of CKD (50 kg/m$^3$), SF (10, 5, and 2.5%), and 1 vol.% of STF and PPF at the curing times of 3, 7, and 28 days.

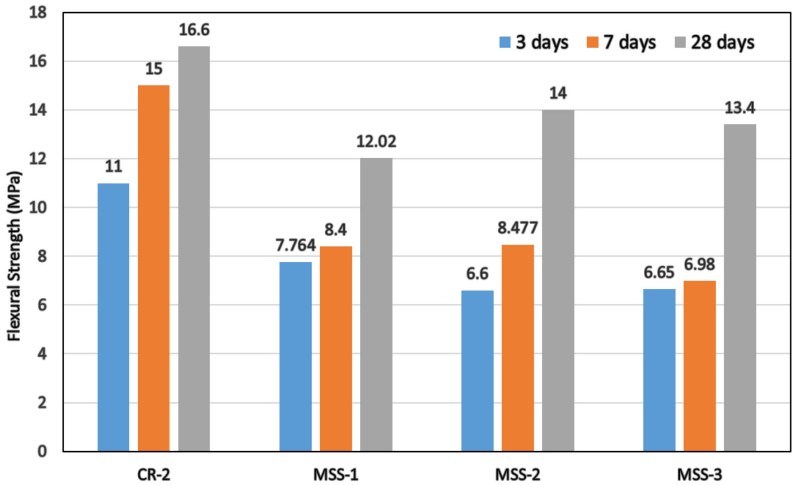

**Figure 19.** Flexural strengths of the mixtures containing partial replacements of MK (10, 15, and 20%), silica fume (15, 10, and 5%), and 1 vol.% of STF at the curing times of 3, 7, and 28 days.

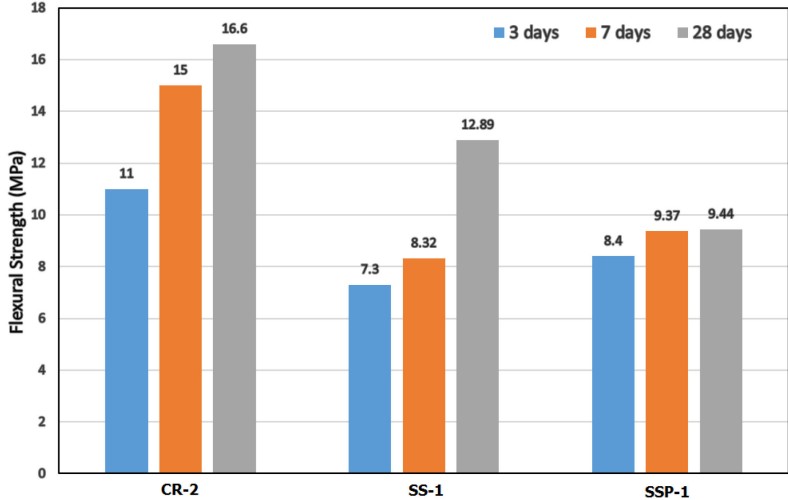

**Figure 20.** Flexural strengths of the mixtures containing partial replacements of a constant SF of 25% and 1 vol.% of STF at curing times of 3, 7, and 28 days.

From an economic point of view, it was observed that the addition of the eco-friendly materials, CKD, SF, and MK, as partial replacements for cement had significant effects on the mechanical properties of the prepared UHPC. Such replacements are a key factor in producing low-cost UHPC and HPC using natural sources. Also, the results indicate that all the prepared UHPC mixtures had excellent flexural strengths compared with the highest control mixture.

Figure 16 illustrates the flexural strengths of the prepared UHPC with different additions of MK. The addition of 100 kg/m$^3$ CKD into the mixtures of the prepared UHPC provided high flexural strengths for all samples. Also, the presence of 1 vol.% of PPF supported the concrete structures to resist high loads. The flexural strengths recorded values of 12.58, 8.68, and 11.34 MPa at 28 d for MK additions of 10, 15, and 20%, respectively. Moreover, sample CR-2 showed a flexural strength of 12.75 MPa at 28 days, the highest value in comparison with other samples. Figure 16 indicates that the flexural strength rose with the increasing MK content. Additionally, the use of MK and CKD makes the preparation of UHPC a more cost-effective method for saving natural resources, thereby protecting the ecological system by reducing environmental pollution. Furthermore, Figure 17 shows the influence of adding both steel and polypropylene fibers on the flexural strengths of the specimens, with steel fiber additions providing a higher flexural strength of the concrete than the concrete with mixed fibers. These results indicated that the use of a hybrid mix of steel and polypropylene fibers in the prepared concrete enhances the flexural behaviors in terms of post-crack strength, toughness ability, and deflection in comparison to other prepared samples. Similarly, Huang et al. [33] found that the length of the steel fibers highly affected the flexural behaviors of UHPC, with the maximum bending strength and toughness achieved with steel fibers of 20 mm in length. The authors noted that the enhancements in the bending behavior and toughness were 19% and 125%, respectively. Jiao et al. [53] also concluded that the presence of hybrid steel fibers of various sizes improves the flexural strength in UHPC. Figures 18–20 show the importance of green additions in the production of UHPCs with high flexural strengths.

### 3.3. Split Tensile Strength

The evaluation of the effect of the splitting load on the prepared cylinders of UHPC provides a clear understanding of the ability of these cylinders to potentially serve under high loads. The amount of cement was reduced in the prepared mixtures to the following amounts: 1000, 750, 500, and 250 kg/m$^3$. The substitution materials were MK (i.e., 10, 15, 40, and 65%) and SF (i.e., 10, 10, 10, and 15%), respectively. All these mixes contained steel fibers of 1% vol. Figure 21 shows the relationships between the splitting tensile strengths of different mixtures of concrete at various curing times. The highest splitting tensile strength of 9.6 MPa at 28 d was noted for the MSS1000 sample, which consisted of 15% MK and a cement content of 1000 kg/m$^3$. Moreover, the reduction in the cement content caused obvious decreases in the splitting tensile strength of other samples, as shown in Figure 21. The high splitting tensile strength in this sample can be attributed to the high cement content as well as to the presence of MK (10%) and SF (10%), with a uniform distributions in the concrete structure.

Also, it was noted that the split tensile strength increased with time, reaching its maximum value at 28 days. Furthermore, the action of the combined MK and SF additions was better at a cement content of 1000 kg/m$^3$ than 750 kg/m$^3$. However, the action of the steel fibers enhanced the splitting tensile strength more than the compressive strength. Adding MK and SF to concrete mixes had a beneficial impact, where substituting them minimized the pore size and strengthened the binding between the silica fume and mortar in the transitional contact zone. Additionally, the prepared UHPC and HPC can be applied successfully to construction requirements with a significant reduction in the amount of cement used by replacing it with eco-friendly materials. Accordingly, the reduction in the cement content provided high-performance concrete with a compressive strength exceeding 120 MPa. All other prepared mixtures of concrete remained within the limits of high-

performance concrete (50–115 MPa). This concrete type is the main construction material in many buildings, bridges, and special applications that must be able to accommodate high loads and stresses [44,53,56].

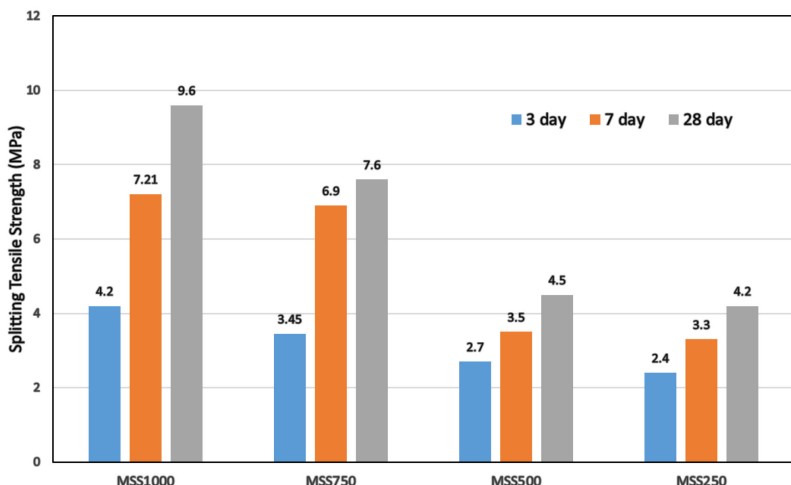

**Figure 21.** Split tensile strengths of the mixtures containing cement of 1000, 750, 500, and 250 kg/m$^3$, MK amounts of 10, 15, 40, and 65%, an SF content of 10, 10, 10, and 15%, and STF of 1% vol. at the curing times of 3, 7, and 28 days.

## 4. Conclusions

UHPC and HPC mixtures were prepared successfully in the present work using eco-friendly materials. The use of MK, SF, CDK, STF, and PPF efficiently enhanced the mechanical properties of concrete. The pretreatment processes employing green additives were key factors in the quality of the final concrete products. The addition of 10% MK, 10% CKD, and 1 vol.% of STF provided the highest compressive strength of 135 MPa. Also, the highest fractural strength was achieved at 19.14 MPa with the additions of MK (15%), CKD (100 kg/m$^3$), and STF (1 vol.%). With the reductions in the cement amount from 1000 to 750, 500, and 250 kg/m$^3$, it was found that the specifications of the prepared UHPC and HPC are still high. Thus, the added green materials managed successfully to replace the cement in the concrete mixture. Such replacement will contribute efficiently to lowering the high economic cost and remediating the environmental problems in the production of classical UHPC. Moreover, minimizing the cement content in the produced concrete resulted in a high-performance concrete with a high compressive strength exceeding 120 MPa. In addition, the splitting tensile strength tests on the concrete demonstrated excellent results. Therefore, such concrete is recommended for use in beams and columns with long spaces due to its high ability to resist various types of shear stresses. Additionally, it is highly recommended to use the prepared UHPC in special structural applications that need high strengths. Highway bridges, rehabilitations, seismic retrofits, and special structures are the main sites that are highly recommended for the use of UHPC. Finally, the prepared concrete is characterized by its high mechanical specifications, by its reduced cement amount, and for containing cheap eco-friendly materials.

**Author Contributions:** Conceptualization, M.F.; methodology, M.Q.A.S.; formal analysis, K.A.S. and M.Q.A.S.; investigation, M.Q.A.S.; data curation, M.Q.A.S.; writing—original draft preparation, M.Q.A.S.; writing (review and editing), K.A.S.; visualization, M.F.; supervision, M.F. and K.A.S.; project administration, M.F. and K.A.S. All authors have read and agreed to the published version of the manuscript.

**Funding:** This research received no external funding.

**Data Availability Statement:** Not applicable.

**Acknowledgments:** The authors are thankful to the Faculty of Civil Engineering, University of Tabriz, Iran, and the Department of Chemical Engineering, University of Technology, Iraq, for their scientific support of this work.

**Conflicts of Interest:** The authors declare no conflict of interest.

## Abbreviations

| Symbol | Description |
|--------|-------------|
| B | binder |
| C | cement |
| CKD | cement kiln dust |
| MK | metakaolin |
| MSD | micro-sand |
| PPF | polypropylene fibers |
| SF | silica fume |
| SP | superplasticizer |
| STF | steel fibers |
| W | water |

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
