# Peer review of "Development and Performance Evaluation of UHPC and HPC Using Eco-Friendly Additions as Substitute Cementitious Materials with Low Cost"

_buildings, doi:10.3390/buildings13082078_

Round 1

Reviewer 1 Report

This paper aims to develop eco-friendly UHPC at a low cost.  There is a major flaw in the work with the claim that the investigation is dealing with UHPC. The mixture design in Table 7 shows that the water-to-binder ratio of the mixtures is 0.32. Typical UHPC mixtures have water-to-binder ratios of less than 0.25 and in many cases lower than 0.22.  Thus, it is difficult to believe that such mixtures can achieve compressive strength as high as 135 MPa at 28 days. Furthermore, from the results of compressive strength, it is shown that the use of the mineral additives results in the reduction of compressive strength to values less than 120 MPa. UHPC should have a minimum compressive strength of 120 MPa at 28 days and in many other standards even higher than that, such as 150 MPa. The title of this manuscript should therefore be changed for this manuscript to be considered.

The paper needs a review for technical writing quality. Please use standard terms, SCM are not "green additives".

Author Response

Dear Reviewer 1,

Thank you very much for your effort in reviewing our manuscript. Fortunately, we have corrected the manuscript according to your useful comments. All the items that you mentioned are corrected and can be seen as blue text in the manuscript to facilitate the review process. Finally, we highly appreciate your notes as they allowed us to appropriately upgrade our manuscript.

Best regards

Reviewer 2 Report

The manuscript deals with the use of alternative ecological materials in concrete. 

The manuscript should be improved based on these requirements.

Line 16: ,,fume silica,, change to ,, silica fume,,

For metakaolin can be used abbreviation MK and for silica fume SF

This manuscript should be discussed in the introduction. https://doi.org/10.3390/ma14112786

The language of the manuscript should be improved.

The fundamental novelty of the work should be stated in the introduction.

The most important abbreviations in the manuscript should be listed.

Add standard deviations to all tables.

Conclusions: List future perspectives.

Moderate editing of English language required.

Author Response

Dear Reviewer 2,

Thank you very much for your effort in reviewing our manuscript. Fortunately, we have corrected the manuscript according to your useful comments. All the items that you mentioned are corrected and can be seen as red text in the manuscript to facilitate the review process. Finally, we highly appreciate your notes as they allowed us to appropriately upgrade our manuscript.

Best regards

Round 2

Reviewer 2 Report

Comments have been resolved, manuscript can be accepted.

Author Response

Thank you so much for the scientific notes.

Best regards
